# WHAT MATTERS FOR ON-POLICY DEEP ACTOR-CRITIC METHODS? A LARGE-SCALE STUDY

**Marcin Andrychowicz, Anton Raichuk, Piotr Stańczyk, Manu Orsini,
Sertan Girgin, Raphaël Marinier, Léonard Hussenot, Matthieu Geist,
Olivier Pietquin, Marcin Michalski, Sylvain Gelly, Olivier Bachem**

Google Research, Brain Team

## ABSTRACT

In recent years, reinforcement learning (RL) has been successfully applied to many different continuous control tasks. While RL algorithms are often conceptually simple, their state-of-the-art implementations take numerous low- and high-level design decisions that strongly affect the performance of the resulting agents. Those choices are usually not extensively discussed in the literature, leading to discrepancy between published descriptions of algorithms and their implementations. This makes it hard to attribute progress in RL and slows down overall progress [27]. As a step towards filling that gap, we implement >50 such "choices" in a unified on-policy deep actor-critic framework, allowing us to investigate their impact in a large-scale empirical study. We train over 250'000 agents in five continuous control environments of different complexity and provide insights and practical recommendations for the training of on-policy deep actor-critic RL agents.

## 1 INTRODUCTION

Deep reinforcement learning (RL) has seen increased interest in recent years due to its ability to have neural-network-based agents learn to act in environments through interactions. For continuous control tasks, on-policy algorithms such as REINFORCE [2], TRPO [10], A3C [14], PPO [17] and off-policy algorithms such as DDPG [13] and SAC [21] have enabled successful applications such as quadrupedal locomotion [20], self-driving [30] or dexterous in-hand manipulation [20, 25, 32].

Many of these papers investigate in depth different algorithmic ideas, for example different loss functions and learning paradigms. Yet, it is less visible that behind successful experiments in deep RL there are complicated code bases that contain a large number of low- and high-level design decisions that are usually not discussed in research papers. While one may assume that such "choices" do not matter, there is evidence that they are in fact crucial for or even driving good performance [27].

While there are open-source implementations available that can be used by practitioners, this is still unsatisfactory: Research publications often contain one-to-one comparisons of different algorithmic ideas based on implementations in different code bases. This makes it impossible to assess whether improvements are due to the underlying algorithmic idea or due to the implementation. In fact, it is hard to assess the performance of high-level algorithmic ideas without an understanding of lower-level choices as performance may strongly depend on the tuning of hyperparameters and implementation-level details. Overall, this makes it hard to attribute progress in reinforcement learning and slows down further research [15, 22, 27].

**Our contributions.** Our key goal in this paper is to investigate such lower level choices in depth and to understand their impact on final agent performance. Hence, as our key contributions, we (1) implement >50 choices in a unified on-policy deep actor-critic implementation[1], (2) conducted a large-scale (more than 250'000 agents trained) experimental study that covers different aspects of the training process, and (3) analyze the experimental results to provide practical insights and recommendations for the training of on-policy deep actor-critic RL agents.

---

[1]The implementation is available at `https://github.com/google-research/seed_rl`.

**Most surprising finding.** While many of our experimental findings confirm common RL practices, some of them are quite surprising, e.g. the policy initialization scheme significantly influences the performance while it is rarely even mentioned in RL publications. In particular, we have found that initializing the network so that the initial action distribution has zero mean, a rather low standard deviation and is independent of the observation significantly improves the training speed (Sec. 3.2).

**Paper outline.** The rest of of this paper is structured as follows: We describe our experimental setup and performance metrics used in Sec. 2. Then, in Sec. 3 we present and analyse the experimental results and finish with related work in Sec. 4 and conclusions in Sec. 5. The appendices contain the detailed description of all design choices we experiment with (App. B), default hyperparameters (App. C) and the raw experimental results (App. D - K).

## 2  STUDY DESIGN

**Considered setting.**   In this paper, we consider the setting of *on-policy deep actor-critic reinforcement learning for continuous control*. We define on-policy learning in the following loose sense: We consider policy iteration algorithms that iterate between generating experience using the current policy and using that experience to improve the policy. This is the standard *modus operandi* of algorithms usually considered on-policy such as PPO [17]. However, we note that algorithms often perform several model updates and thus may operate technically on off-policy data within a single policy improvement iteration. As benchmark environments, we consider five widely used continuous control environments from OpenAI Gym [12] of varying complexity: Hopper-v1, Walker2d-v1, HalfCheetah-v1, Ant-v1, and Humanoid-v1 [2].

**Unified on-policy deep actor-critic gradient algorithm.**   We took the following approach to create a highly configurable unified on-policy deep actor-critic gradient algorithm with as many choices as possible:

1. We researched prior work and popular code bases to make a list of commonly used choices, i.e., different loss functions (both for value functions and policies), architectural choices such as initialization methods, heuristic tricks such as gradient clipping and all their corresponding hyperparameters.
2. Based on this, we implemented a single, unified on-policy deep actor-critic agent and corresponding training protocol starting from the SEED RL code base [28]. Whenever we were faced with implementation decisions that required us to take decisions that could not be clearly motivated or had alternative solutions, we further added such decisions as additional choices.
3. We verified that when all choices are selected as in the PPO implementation from OpenAI baselines, we obtain similar performance as reported in the PPO paper [17]. We chose PPO because it is probably the most commonly used on-policy deep actor-critic RL algorithm at the moment.

The resulting agent implementation is detailed in Appendix B. The key property is that the implementation exposes all choices as configuration options in an unified manner. For convenience, we mark each of the choice in this paper with a number (e.g., `C1`) and a fixed name (e.g. `num_envs (C1)`) that can be easily used to find a description of the choice in Appendix B.

**Difficulty of investigating choices.**   The primary goal of this paper is to understand how the different choices affect the final performance of an agent and to derive recommendations for these choices. There are two key reasons why this is challenging:

First, we are mainly interested in insights on choices for good hyperparameter configurations. Yet, if all choices are sampled randomly, the performance is very bad and little (if any) training progress is made. This may be explained by the presence of sub-optimal settings (e.g., hyperparameters of the wrong scale) that prohibit learning at all. If there are many choices, the probability of such failure increases exponentially.

---

[2]It has been noticed that the version of the Mujoco physics simulator [5] can slightly influence the behaviour of some of the environments — `https://github.com/openai/gym/issues/1541`. We used Mujoco 2.0 in our experiments.

Second, many choices may have strong interactions with other related choices, for example the learning rate and the minibatch size. This means that such choices need to be tuned together and experiments where only a single choice is varied but interacting choices are kept fixed may lead to misleading conclusions.

**Basic experimental design.** To address these issues, we design a series of experiments as follows: We create groups of choices around thematic groups where we suspect interactions between different choices, for example we group together all choices related to neural network architecture. We also include `Adam learning rate (C24)` in all of the groups as we suspect that it may interact with many other choices.

Then, in each experiment, we train a large number of models where we randomly sample the choices within the corresponding group [3]. All other settings (for choices not in the group) are set to settings of a competitive base configuration (detailed in Appendix C) that is close to the default PPOv2 configuration[4] scaled up to 256 parallel environments. This has two effects: First, it ensures that our set of trained models contains good models (as verified by performance statistics in the corresponding results). Second, it guarantees that we have models that have different combinations of potentially interacting choices.

We consider two different analyses for each choice (e.g, for `advantage_estimator (C6)`):

*Conditional 95th percentile*: For each potential value of that choice (e.g., `advantage_estimator (C6) = N-Step`), we look at the performance distribution of sampled configurations with that value. We report the 95th percentile of the performance as well as a confidence interval based on a binomial approximation [5]. Intuitively, this corresponds to a robust estimate of the performance one can expect if all other choices in the group were tuned with random search and a limited budget of roughly 20 hyperparameter configurations.

*Distribution of choice within top 5% configurations.* We further consider for each choice the distribution of values among the top 5% configurations trained in that experiment. The reasoning is as follows: By design of the experiment, values for each choice are distributed uniformly at random. Thus, if certain values are over-represented in the top models, this indicates that the specific choice is important in guaranteeing good performance.

**Performance measures.** We employ the following way to compute performance: For each choice configuration, we train 3 models with independent random seeds where each model is trained for one million (Hopper, HalfCheetah, Walker2d) or two million environment steps (Ant, Humanoid). We evaluate trained policies every hundred thousand steps by freezing the policy and computing the average undiscounted episode return of 100 episodes (with the stochastic policy). We then average these score to obtain a single performance score of the seed which is proportional to the area under the learning curve. This ensures we assign higher scores to agents that learn quickly. The performance score of a hyperparameter configuration is finally set to the median performance score across the 3 seeds. This reduces the impact of training noise, i.e., that certain seeds of the same configuration may train much better than others.

**Robustness of results.** While we take 3 random seeds to compute the performance measure for a single choice configuration, it is important to note that all the experimental results reported in this paper are based on more than 3 random seeds: The reported *conditional 95th percentile* and *distribution of choice within top 5% configurations* are computed based upon the performance of hundreds of choice configurations. Furthermore, we also report confidence intervals for the *conditional 95th percentile*.

---

[3] Exact details for the different experiments are provided in Appendices D - K.

[4] https://github.com/openai/baselines/blob/master/baselines/ppo2/defaults.py

[5] We compute confidence intervals with a significance level of $\alpha = 5\%$ as follows: We find $i_l = \text{icdf}\left(\frac{\alpha}{2}\right)$ and $i_h = \text{icdf}\left(1 - \frac{\alpha}{2}\right)$ where icdf is the inverse cumulative density function of a binomial distribution with $p = 0.95$ (as we consider the 95th percentile) and the number of draws equals the number of samples. We then report the $i_l$th and $i_h$th highest scores as the confidence interval.

## 3 EXPERIMENTS

We run experiments for eight thematic groups: *Policy Losses* (Sec. 3.1), *Networks architecture* (Sec. 3.2), *Normalization and clipping* (Sec. 3.3), *Advantage Estimation* (Sec. 3.4), *Training setup* (Sec. 3.5), *Timesteps handling* (Sec. 3.6), *Optimizers* (Sec. 3.7), and *Regularization* (Sec. 3.8). For each group, we provide a full experimental design and full experimental plots in Appendices D - K so that the reader can draw their own conclusions from the experimental results. Moreover, the raw data from all training runs and a script used to generate all plots for this paper can be found online[6]. In the following sections, we provide short descriptions of the experiments, our interpretation of the results, as well as practical recommendations for agent training for continuous control.

### 3.1 POLICY LOSSES (BASED ON THE RESULTS IN APPENDIX D)

**Study description.** We investigate different policy losses (C14): vanilla policy gradient (PG), V-trace [19], PPO [17], AWR [33], V-MPO[7] [34] and the limiting case of AWR ($\beta \to 0$) and V-MPO ($\epsilon_n \to 0$) which we call Repeat Positive Advantages (RPA) as it is equivalent to the negative log-probability of actions with positive advantages. See App. B.3 for a detailed description of the different losses. We further sweep the hyperparameters of each of the losses (C15, C16, C18, C17, C19), the learning rate (C24) and the number of passes over the data (C3).

The goal of this study is to better understand the importance of the policy loss function in the on-policy deep actor-critic setting considered in this paper. The goal is **not** to provide a general statement that one of the losses is better than the others as some of them were specifically designed for other settings (e.g., the V-trace loss is targeted at near-on-policy data in a distributed setting).

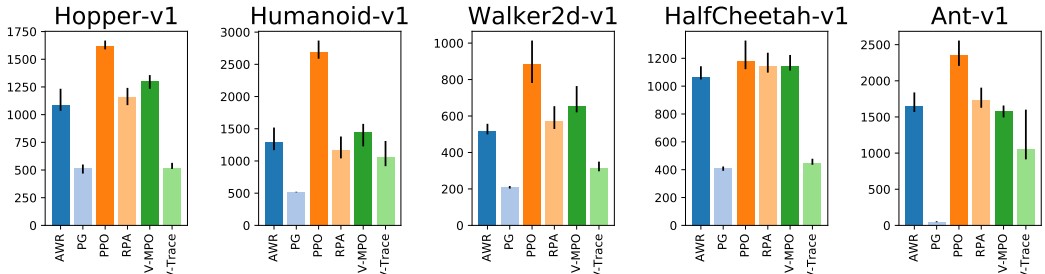

Figure 1: Comparison of different policy losses (C14).

**Interpretation.** Fig. 1 shows the 95-th percentile of the average policy score during training for different policy losses (C14). We observe that PPO performs better than the other losses on 4 out of 5 environments and is one of the top performing losses on HalfCheetah. As we randomly sample the loss specific hyperparameters in this analysis, one might argue that our approach favours choices that are not too sensitive to hyperparameters. At the same time, there might be losses that are sensitive to their hyperparameters but for which good settings may be easily found. Fig. 5 shows that even if we condition on choosing the optimal loss hyperparameters for each loss[8], PPO still outperforms the other losses on the two hardest tasks — Humanoid and Ant[9] and is one of the top performing losses on the other 3 tasks. Moreover, we show the empirical cumulative density functions of agent performance conditioned on the policy loss used in Fig. 4.

Perhaps unsurprisingly, PG and V-trace perform worse on all tasks. This is likely caused by their inability to handle data that becomes off-policy in one iteration, either due to multiple passes (C3)

---

[6]https://github.com/google-research/seed_rl/blob/master/mujoco/what_matters_in_on_policy_rl.ipynb

[7]We used the V-MPO policy loss without the decoupled KL constraint as we investigate the effects of different policy regularizers separately in Sec. 3.8.

[8]AWR loss has two hyperparameters — the temperature $\beta$ (C18) and the weight clipping coefficient $\omega_{max}$ (C17). We only condition on $\beta$ which is more important.

[9]These two tasks were not included in the original PPO paper [17] so the hyperparameters we use were not tuned for them.

over experience (which can be seen in Fig. 14) or a large experience buffer (C2) in relation to the batch size (C4). While V-Trace contains an off-policy correction, it was designed for "slightly" off-policy experience arising in ansynchronous, distributed setups and becomes more and more biased as experience becomes more off-policy. Overall, these results show that trust-region optimization (preventing the current policy from diverging too much from the behavioral one) which is present in all the other policy losses is crucial for good sample complexity.

For PPO and its clipping threshold $\epsilon$ (C16), we further observe that $\epsilon = 0.2$ and $\epsilon = 0.3$ perform reasonably well in all environments but that lower ($\epsilon = 0.1$) or higher ($\epsilon = 0.5$) values give better performance on some of the environments (See Fig. 10 and Fig. 32).

**Recommendation.** Use the PPO policy loss. Start with the clipping threshold set to $0.25$ but also try lower and higher values if possible.

### 3.2 NETWORKS ARCHITECTURE (BASED ON THE RESULTS IN APPENDIX E)

**Study description.** We investigate the impact of differences in the policy and value function neural network architectures. We consider choices related to the network structure and size (C47, C48, C49, C50, C51, C52, C52), activation functions (C55), and initialization of network weights (C56, C57, C58). We further include choices related to the standard deviation of actions (C59, C60, C61, C62) and transformations of sampled actions (C63).

**Interpretation.** Separate value and policy networks (C47) appear to lead to better performance on four out of five environments (Fig. 15). To avoid analyzing the other choices based on bad models, we thus focus for the rest of this experiment only on agents with separate value and policy networks. Regarding network sizes, the optimal width of the policy MLP depends on the complexity of the environment (Fig. 18) and too low or too high values can cause significant drop in performance while for the value function there seems to be no downside in using wider networks (Fig. 21). Moreover, on some environments it is beneficial to make the value network wider than the policy one, e.g. on HalfCheetah the best results are achieved with $16 - 32$ units per layer in the policy network and $256$ in the value network. Two hidden layers appear to work well for policy (Fig. 22) and value networks (Fig. 20) in all tested environments. As for activation functions, we observe that tanh activations perform best and relu worst. (Fig. 30).

Interestingly, the initial policy appears to have a surprisingly high impact on the training performance. The key recipe is to initialize the policy at the beginning of training so that the action distribution is centered around $0$[10] regardless of the observation and has a rather small standard deviation. This can be achieved by initializing the policy MLP with smaller weights in the last layer (C57, Fig. 24, this alone boosts the performance on Humanoid by 66%) so that the initial action distribution is almost independent of the observation and by introducing an offset in the standard deviation of actions (C61). Fig. 2 shows that the performance is very sensitive to the initial action standard deviation with 0.5 performing best on all environments except Hopper where higher values perform better.

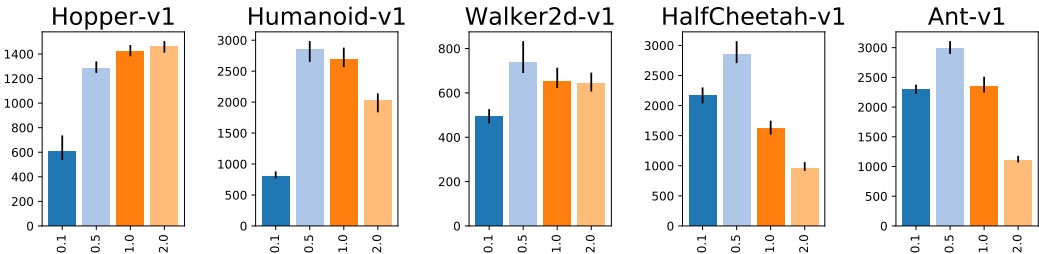

Figure 2: Comparison of different initial standard deviations of actions (C61).

Fig. 17 compares two approaches to transform unbounded sampled actions into the bounded $[-1, 1]$ domain expected by the environment (C63): clipping and applying a tanh function. tanh performs slightly better overall (in particular it improves the performance on HalfCheetah by 30%). Comparing

---

[10]All environments expect normalized actions in $[-1, 1]$.

Fig. 17 and Fig. 2 suggests that the difference might be mostly caused by the decreased magnitude of initial actions[11].

Other choices appear to be less important: The scale of the last layer initialization matters much less for the value MLP (`C58`) than for the policy MLP (Fig. 19). Apart from the last layer scaling, the network initialization scheme (`C56`) does not matter too much (Fig. 27). Only `he_normal` and `he_uniform` [7] appear to be suboptimal choices with the other options performing very similarly. There also appears to be no clear benefits if the standard deviation of the policy is learned for each state (i.e. outputted by the policy network) or once globally for all states (`C59`, Fig. 23). For the transformation of policy output into action standard deviation (`C60`), `softplus` and exponentiation perform very similarly[12] (Fig. 25). Finally, the minimum action standard deviation (`C62`) seems to matter little, if it is not set too large (Fig. 30).

**Recommendation.** Initialize the last policy layer with $100\times$ smaller weights. Use `softplus` to transform network output into action standard deviation and add a (negative) offset to its input to decrease the initial standard deviation of actions. Tune this offset if possible. Use `tanh` both as the activation function (if the networks are not too deep) and to transform the samples from the normal distribution to the bounded action space. Use a wide value MLP (no layers shared with the policy) but tune the policy width (it might need to be narrower than the value MLP).

## 3.3 NORMALIZATION AND CLIPPING (BASED ON THE RESULTS IN APPENDIX F)

**Study description.** We investigate the impact of different normalization techniques: observation normalization (`C64`), value function normalization (`C66`), per-minibatch advantage normalization (`C67`), as well as gradient (`C68`) and observation (`C65`) clipping.

**Interpretation.** Input normalization (`C64`) is crucial for good performance on all environments apart from Hopper (Fig. 33). Quite surprisingly, value function normalization (`C66`) also influences the performance very strongly — it is crucial for good performance on HalfCheetah and Humanoid, helps slightly on Hopper and Ant and significantly *hurts* the performance on Walker2d (Fig. 37). We are not sure why the value function scale matters that much but suspect that it affects the performance by changing the speed of the value function fitting.[13] In contrast to observation and value function normalization, per-minibatch advantage normalization (`C67`) seems not to affect the performance too much (Fig. 35). Similarly, we have found little evidence that clipping normalized[14] observations (`C65`) helps (Fig. 38) but it might be worth using if there is a risk of extremely high observations due to simulator divergence. Finally, gradient clipping (`C68`) provides a small performance boost with the exact clipping threshold making little difference (Fig. 34).

**Recommendation.** Always use observation normalization and check if value function normalization improves performance. Gradient clipping might slightly help but is of secondary importance.

## 3.4 ADVANTAGE ESTIMATION (BASED ON THE RESULTS IN APPENDIX G)

**Study description.** We compare the most commonly used advantage estimators (`C6`): N-step [3], GAE [9] and V-trace [19] and their hyperparameters (`C7`, `C8`, `C9`, `C10`). We also experiment with applying PPO-style pessimistic clipping (`C13`) to the value loss (present in the original PPO implementation but not mentioned in the PPO paper [17]) and using Huber loss [1] instead of MSE for value learning (`C11`, `C12`). Moreover, we varied the number of parallel environments used (`C1`) as it changes the length of the experience fragments collected in each step.

**Interpretation.** GAE and V-trace appear to perform better than N-step returns (Fig. 44 and 40) which indicates that it is beneficial to combine the value estimators from multiple timesteps. We have not

---

[11]`tanh` can also potentially perform better with entropy regularization (not used in this experiment) as it bounds the maximum possible policy entropy.

[12]We noticed that some of the training runs with exponentiation resulted in NaNs but clipping the exponent solves this issue (See Sec. B.8 for the details).

[13]Another explanation could be the interaction between the value function normalization and PPO-style value clipping (`C13`). We have, however, disabled the value clipping in this experiment to avoid this interaction. The disabling of the value clipping could also explain why our conclusions are different from [27] where a form of value normalization improved the performance on Walker.

[14]We only applied clipping if input normalization was enabled.

found a significant performance difference between GAE and V-trace in our experiments. $\lambda = 0.9$ (`C8`, `C9`) performed well regardless of whether GAE (Fig. 45) or V-trace (Fig. 49) was used on all tasks but tuning this value per environment may lead to modest performance gains. We have found that PPO-style value loss clipping (`C13`) hurts the performance regardless of the clipping threshold[15] (Fig. 43). Similarly, the Huber loss (`C11`) performed worse than MSE in all environments (Fig. 42) regardless of the value of the threshold (`C12`) used (Fig. 48).

**Recommendation.** Use GAE with $\lambda = 0.9$ but neither Huber loss nor PPO-style value loss clipping.

### 3.5 TRAINING SETUP (BASED ON THE RESULTS IN APPENDIX H)

**Study description.** We investigate choices related to the data collection and minibatch handling: the number of parallel environments used (`C1`), the number of transitions gathered in each iteration (`C2`), the number of passes over the data (`C3`), minibatch size (`C4`) and how the data is split into minibatches (`C5`).

For the last choice, in addition to standard choices, we also consider a new small modification of the original PPO approach: The original PPO implementation splits the data in each policy iteration step into individual transitions and then randomly assigns them to minibatches (`C5`). This makes it impossible to compute advantages as the temporal structure is broken. Therefore, the advantages are computed once at the beginning of each policy iteration step and then used in minibatch policy and value function optimization. This results in higher diversity of data in each minibatch at the cost of using slightly stale advantage estimations. As a remedy to this problem, we propose to recompute the advantages at the beginning of each pass over the data instead of just once per iteration.

**Results.** Unsurprisingly, going over the experience multiple times appears to be crucial for good sample complexity (Fig. 54). Often, this is computationally cheap due to the simple models considered, in particular on machines with accelerators such as GPUs and TPUs. As we increase the number of parallel environments (`C1`), performance decreases sharply on some of the environments (Fig. 55). This is likely caused by shortened experience chunks (See Sec. B.1 for the detailed description of the data collection process) and earlier value bootstrapping. Despite that, training with more environments usually leads to faster training in wall-clock time if enough CPU cores are available. Increasing the batch size (`C4`) does not appear to hurt the sample complexity in the range we tested (Fig. 57) which suggests that it should be increased for faster iteration speed. On the other hand, the number of transitions gathered in each iteration (`C2`) influences the performance quite significantly (Fig. 52). Finally, we compare different ways to handle minibatches (See App. B.1 for the detailed description of different variants) in Fig. 53 and 58. The plots suggest that stale advantages can in fact hurt performance and that recomputing them at the beginning of each pass at least partially mitigates the problem and performs best among all variants.

**Recommendation.** Go over experience multiple times. Shuffle individual transitions before assigning them to minibatches and recompute advantages once per data pass (See App. B.1 for the details). For faster wall-clock time training use many parallel environments and increase the batch size (both might hurt the sample complexity). Tune the number of transitions in each iteration (`C2`) if possible.

### 3.6 TIMESTEPS HANDLING (BASED ON THE RESULTS IN APPENDIX I)

**Study description.** We investigate choices related to the handling of timesteps: discount factor[16] (`C20`), frame skip (`C21`), and how episode termination due to timestep limits are handled (`C22`). The latter relates to a technical difficulty explained in App. B.4 where one assumes for the algorithm an infinite time horizon but then trains using a finite time horizon [16].

**Interpretation.** Fig. 60 shows that the performance depends heavily on the discount factor $\gamma$ (`C20`) with $\gamma = 0.99$ performing reasonably well in all environments. Skipping every other frame (`C21`) improves the performance on 2 out of 5 environments (Fig. 61). Proper handling of episodes abandoned due to the timestep limit seems not to affect the performance (`C22`, Fig. 62) which

---

[15]This is consistent with prior work [27].

[16]While the discount factor is sometimes treated as a part of the environment, we assume that the real goal is to maximize *undiscounted* returns and the discount factor is a part of the algorithm which makes learning easier.

is probably caused by the fact that the timestep limit is quite high (1000 transitions) in all the environments we considered.

**Recommendation.** Discount factor $\gamma$ is one of the most important hyperparameters and should be tuned per environment (start with $\gamma = 0.99$). Try frame skip if possible. There is no need to handle environments step limits in a special way for large step limits.

### 3.7 OPTIMIZERS (BASED ON THE RESULTS IN APPENDIX J)

**Study description.** We investigate two gradient-based optimizers commonly used in RL: (`C23`) – Adam [8] and RMSprop – as well as their hyperparameters (`C24`, `C25`, `C26`, `C27`, `C28`, `C29`, `C30`) and a linear learning rate decay schedule (`C31`).

**Interpretation.** The differences in performance between the optimizers (`C23`) appear to be rather small with no optimizer consistently outperforming the other across environments (Fig. 66). Unsurprisingly, the learning rate influences the performance very strongly (Fig. 69) with the default value of 0.0003 for Adam (`C24`) performing well on all tasks. Fig. 67 shows that Adam works better with momentum (`C26`). For RMSprop, momentum (`C27`) makes less difference (Fig. 71) but our results suggest that it might slightly improve performance[17]. Whether the centered or uncentered version of RMSprop is used (`C30`) makes no difference (Fig. 70) and similarly we did not find any difference between different values of the $\epsilon$ coefficients (`C28`, `C29`, Fig. 68 and 72). Linearly decaying the learning rate to 0 increases the performance on 4 out of 5 tasks but the gains are very small apart from Ant, where it leads to 15% higher scores (Fig. 65).

**Recommendation.** Use Adam [8] optimizer with momentum $\beta_1 = 0.9$ and a tuned learning rate (0.0003 is a safe default). Linearly decaying the learning rate may slightly improve performance but is of secondary importance.

### 3.8 REGULARIZATION (BASED ON THE RESULTS IN APPENDIX K)

**Study description.** We investigate different policy regularizers (`C32`), which can have either the form of a penalty (`C33`, e.g. bonus for higher entropy) or a soft constraint (`C34`, e.g. entropy should not be lower than some threshold) which is enforced with a Lagrange multiplier. In particular, we consider the following regularization terms: entropy (`C40`, `C46`), the Kullback–Leibler divergence (KL) between a reference $\mathcal{N}(0, 1)$ action distribution and the current policy (`C37`, `C43`) and the KL divergence and reverse KL divergence between the current policy and the behavioral one (`C35`, `C41`, `C36`, `C42`), as well as the "decoupled" KL divergence from [18, 34] (`C38`, `C39`, `C44`, `C45`).

**Interpretation.** We do not find evidence that regularization helps significantly on our environments with the exception of HalfCheetah on which all constraints (especially the entropy constraint) help (Fig. 76 and 77). However, the performance boost is largely independent on the constraint threshold (Fig. 83, 84, 87, 89, 90 and 91) which suggests that the effect is caused by the initial high strength of the penalty (before it gets adjusted) and not by the desired constraint. While it is surprising that regularization does not help at all (apart from HalfCheetah), we conjecture that regularization might be less important in our experiments because: (1) the PPO policy loss already enforces the trust region which makes KL penalties or constraints redundant; and (2) the careful policy initialization (See Sec. 3.2) is enough to guarantee good exploration making the entropy bonus or constraint redundant.

## 4 RELATED WORK

Islam et al. [15] and Henderson et al. [22] point out reproducibility issues in RL including the performance differences between different code bases, the importance of hyperparameter tuning and the high level of stochasticity due to random seeds. Tucker et al. [26] showed that the gains, which had been attributed to one of the recently proposed policy gradients improvements, were, in fact, caused by the implementation details. The most closely related work to ours is probably Engstrom et al. [27] where the authors investigate code-level improvements in the PPO [17] code base and conclude that they are responsible for the most of the performance difference between PPO

---

[17]Importantly, switching from no momentum to momentum 0.9 increases the RMSprop step size by approximately $10\times$ and requires an appropriate adjustment to the learning rate (Fig. 74).

and TRPO [10]. Our work is also similar to other large-scale studies done in other fields of Deep Learning, e.g. model-based RL [31], GANs [24], NLP [35], disentangled representations [23] and convolution network architectures [36].

## 5  CONCLUSIONS

In this paper, we investigated the importance of a broad set of high- and low-level choices that need to be made when designing and implementing on-policy deep actor-critic RL algorithms. Based on more than 250'000 experiments in five continuous control environments, we evaluate the impact of different choices and provide practical recommendations. One of the surprising insights is that the initial action distribution plays an important role in agent performance. We expect this to be a fruitful avenue for future research.

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
