# OpenReview forum: "What Matters for On-Policy Deep Actor-Critic Methods? A Large-Scale Study"
_ICLR.cc/2021/Conference — ICLR 2021 Oral_

### Official Review · AnonReviewer1 · 2020-10-27
**Empirical evaluation of many algorithmic choices in on-policy DRL with actionable takeaways**

**Rating:** 7
**Confidence:** 4

**Review:**

The paper presents an empirical evaluation of many algorithmic choices made in the implementations of on-policy actor-critic algorithms in deep reinforcement learning (RL). The authors group those choices in clusters in which they expect some interactions. For each cluster, they test sets of randomly made choices while assuming that choices outside a cluster are set to competitive default values. Based on those experimental results, the authors formulate recommendations about how to make those choices for each cluster.


PROS
The paper is well-written and clear. This paper is part of the string of recent papers that discuss the difficulty of evaluating deep RL algorithms. I appreciate the breadth of the choices that the authors consider. The justification for their overall experimental design (i.e., evaluating per choice clusters) is reasonable. While some findings are as expected, others are indeed unexpected and not discussed in the deep RL literature.


CONS
I have a doubt about the robustness of the results. The authors decided to use the median over 3 seeds for the evaluation. Although the median is used, is it reliable given the observations made by Henderson et al., which implies that performance can vary a lot with respect to seeds? Could the authors comment on that point?

I think one important missing experiment is the evaluation of the combinations of all the recommendations made in the paper. Do the recommendations depend on the default setting for other choices, do they have a synergetic effect or could there be some negative interactions?

---

> ### Author Response · Authors · 2020-11-24
> **Authors' response**
>
> We thank the reviewer for their review. Please find our answers below:
>
> 1. Median across 3 random seeds: This was mainly due to computational constraints. Furthermore, while we take the median of independent runs for each configuration, we compute the 95-th percentile of this score across hundreds of configurations (where the choice is set to a specific value) and report confidence intervals. Hence, the numbers reported in the paper are the results of not only three but hundreds of training runs.
>
> 2. Evaluation of recommendations in the paper: We have successfully applied the recommendation from the paper on a variety of different environments and obtained very competitive agents. We will aim to include an experiment with such tangible results for the final manuscript.

---

### Official Review · AnonReviewer3 · 2020-10-28
**An excellent survey of a very hard empirical field**

**Rating:** 9
**Confidence:** 3

**Review:**

The authors survey a wide variety of implementation-level and hyperparameter decisions in reinforcement learning for continuous control tasks. They train over 250.000 agents with different settings and suggest empirical guidelines.

As the authors indicate, there's not too much related work so one could call this work pioneering: The sort of work conducted by the authors is crucially important for a field afloat with tricks and tweaks, many of which are typically not discussed in the scientific literature due to a misplaced conceit around this being "not research, just engineering" entirely absent from established fields of science such as experimental Physics. It is also typically not done as it is just plain hard to do. Combined with the often lackluster response from the community, the cost-benefits trade-off has just not been worth it, especially for junior researchers. It's all the more commendable that the authors have engaged with this formidable task of bringing some of the "secret sauce" out of the heads of senior engineers in the various labs and into published and peer-reviewed science.

The authors compare various choices of configurations obtained from the Cartesian product of 8 factors which they call thematic groups: Policy Losses (Sec. 3.1), Networks architecture (Sec. 3.2), Normalization and clipping (Sec. 3.3), Advantage Estimation (Sec. 3.4), Training setup (Sec. 3.5), Timesteps handling (Sec. 3.6), Optimizers (Sec. 3.7), and Regularization (Sec. 3.8). The high-variance nature of training RL agents makes it such that the individual factors in these configurations often have surprising non-linear cross-relations such that the problem space cannot be evaluated incrementally (i.e., it's often not possible to establish "the best" architecture first and select the right learning rate afterwards). The authors propose a novel approach of considering for each choice the distribution of values among the top 5% configurations trained in that experiment. Their experimental design is such that the values for each choice are distributed uniformly at random and thus if certain values are over-represented in the top models this indicates that the specific choice is important in guaranteeing good performance.

As for improvements on the paper, I have one major and only a few minor comments. My major comment is that the paper does not indicate anywhere that the research code is released, only that it's based on SEED RL. I believe an authoritative public implementation of the configurations considered would be extremely worthwhile, both for the community and the authors. If they haven't already done so (there's no supplements to this submission and I refrained from doing any web searches to preserve anonymity), I'd urge the authors to invest the time to release a (possibly cleaned-up) version of their code.

As for minor comments, I'm not clear about the philosophical distinction of something being "due to the algorithms or due to their implementations" (in the Introduction). I very much see the point the authors are making, which is an important one -- what makes RL results work is often "nitty-gritty" details not mentioned in the main part of the relevant publications (and often just barely mentioned in appendices). However, in the strictest sense, the algorithm very much is the implementation -- that's what produces a given result. It's worthwhile to keep the distinction between an idea (say, PPO) and a given implementation of that idea (e.g., presumably the authors had to re-implement PPO in TF2 when using SEED RL and couldn't use OpenAI's original implementation). It's also fine to call the idea "the algorithm", but I'd have preferred to see this distinction more clearly defined.

Somewhat related: The authors are very much correct about what they call "standard modus operandi of algorithms [...] such as PPO", namely iterating between generating experience using the current policy and using the experience to improve the policy. I'd add that strictly speaking no _iteration_ is necessary, as for instance IMPALA, coming from the A3C line of development, does both asynchronously in parallel, and I suspect so do the authors given their use of SEED RL. My suggestion would be to slightly rephrase this sentence and mention IMPALA along with PPO. Perhaps there could also be a comment somewhere about what constitutes "PPO" (or "IMPALA") -- e.g., IMPALA consists of (1) an asynchronous actor/learner split [with further choices of when/how the weights are copied from learner to actor, see e.g. [this comment](https://github.com/deepmind/scalable_agent/blob/master/experiment.py#L508)], (2) a specific type of policy-gradient loss, v-trace, (3) a specific neural network architecture, optionally including recurrence via an LSTM and potentially even (4) a specific type of preprocessing for environments such as Atari or DMLab [according to some papers, swapping out the implementation of the frame-downscaling algorithm in Atari has measurable impact on final performance -- this will matter a lot when evaluating re-implementations of an "algorithm"]. I'd like for the authors to take on this opportunity and propose a common language to discuss these distinctions, which in practise are often confusing to junior researchers (and some senior researchers, too).

Further related to IMPALA and v-trace, I was surprised about the word "unsurprisingly" and the explanation in "Perhaps unsurprisingly, PG and V-trace perform worse on all tasks. This is likely caused by their inability to handle data that become off-policy in one iteration, either due to multiple passes over experience [...] or a large experience buffer in relation to the batch size." While the results speak for themselves, my understanding of v-trace was that it was specifically designed for the very goal of dealing with the "slight" off-policiness produced by asynchronous actor/learner splits in a PG setting. Perhaps the authors have an intuition I'm lacking at this point, but if so I'd appreciate further elaboration.

As a final and perhaps trivial comment, I was slightly irritated by the notation/typography for the inverse cumulative density function of a binomial distribution. In $\rm\LaTeX$, the symbols $icdf$ read as the in-context nonsensical $i\cdot c\cdot d\cdot f$ while the authors would presumably want to use $\mathrm{icdf}$ (compare $exp(x)$ vs $\exp(x)$ or $sin(x)$ vs $\sin(x)$). I'd propose `\mathrm{icdf}`  as the correct syntax for this in $\rm\LaTeX$.

In follow-up work, I'd like to see a similar paper for various "discrete RL" tasks (a subset of Atari, VizDoom, DMLab, MiniGrid, BabyAI, ProcGen, and perhaps even Obstacle Tower, Minecraft, StarCraft (I or II) or the recent NetHack environment) with similar factors of configurations. I assume this is a task yet more daunting, but no less useful to the overall community of researchers.

Overall, this is a strong paper and I recommend it for publication.

---

> ### Author Response · Authors · 2020-11-24
> **Authors' response.**
>
> We thank the reviewer for their review. Please find our responses below:
>
> 1. Open source implementation: We will release an open source implementation of the agent.
>
> 2. Terminology (idea/algorithm/implementation): We will carefully revise the manuscript to make the distinction between these terms more clear.
>
> 3. "standard modus operandi of algorithms" (IMPALA vs PPO)/different "ingredients" of IMPALA. This is a good point and we will add additional explanations to the manuscript explaining these differences.
>
> 4. "Perhaps unsurprisingly, PG and V-trace perform worse on all tasks". This is based on the following argument/intuition: In a given iteration of the considered setup, the learning algorithm may take 320 gradient steps (10 passes over 2048 transitions with the minibatch size of 64) until new experience is generated. Hence, data may not be "slightly" off-policy as in the IMPALA asynchronous setup. While V-Trace becomes more biased as experience becomes more off-policy, PPO limits how far the policy deviates in a single iteration from the prior policy via an implicit trust region. Hence, it can be expected that this yields different results in the given setup.
>
> 5. Notation/typography for the inverse cumulative density function of a binomial distribution: We will revise the notation based on the reviewer's feedback.

---

### Official Review · AnonReviewer4 · 2020-10-28
**Recommendation to accept**

**Rating:** 9
**Confidence:** 3

**Review:**

##########################################################################

Summary:

This paper conducted a large-scale empirical study that provides insights and practical recommendations for the training of on-policy deep actor-critic RL agents

##########################################################################

Reasons for score:

Overall, I'd vote for acceptance to the paper. The paper is informative and practical; however, I'm not sure that the paper meets ICLR's requirement.

Pros:
	1. Reproducibility is one of the main issues for various RL algorithms. This paper conducts a large-scale empirical study for popular on-policy algorithms.
	2. The paper is well-written, and the suggestion is useful to me.


Sorry, but I didn't go through all the details in the appendix.

---

> ### Author Response · Authors · 2020-11-24
> **Authors' response**
>
> We thank the reviewer for their review.

---

### Official Review · AnonReviewer2 · 2020-10-30
**Solid work, and interesting results.**

**Rating:** 7
**Confidence:** 4

**Review:**

This paper carries out a large-scale study for understanding of on-policy deep actor-critic. The study looks into a large choices of many implementation settings and design decisions, and investigate their impact on the task performance. The evaluations are done with 250000 RL agents on 5 different continuous control tasks. For each evaluation category, there is a finding summary that provides practical recommendations.

In overall, this study is exhaustive and helpful to both RL researchers and practitioners. The experiment organization which separates all design choices into 7 main categories is very excellent in a systematic way. They cover most design choices in recent works of on-policy RL methods. The reports and the interpretation of results are very interesting and easy to read. The main and important findings are summarized concisely and expected to play important hints.


The only performance metric studied in the paper is a score that is proportional to the area under the learning curve. I was wondering if there should be an additional metric, i.e the final policy or an average reward of the final 100 policies? Would the final or best policy be of more interest to the choice of a practitioner?

As many recent work investigates the design choice of only on-policy RL methods, it would be interesting if in introduction there is discussion on why off-policy methods are not considered or should it be addressed in a different way in another research?

Beside the focus on only the performance in terms of rewards, it would be interesting if the discussion can be expanded to look at other matters, e.g. numerical stability of design/hyper-parameter choices, convergence behaviors (it might requires plot to see if a method show premature convergence, fast learning but sub-optimal, fluctuating, etc.).

Although the paper only uses Mujoco simulator, would the hyperparameters' domains be subjective to it, e.g. inertia, fiction, joint limits, contacts, etc.? It would be helpful if the discussion can show if such those factors play any role in the results? It would lead to more helpful finding summary.


As a final comment, this is a solid work and will be very helpful to the community. Given that it is implemented on the new SEED RL framework, so it would be better if the implementation code can be published.

---

> ### Author Response · Authors · 2020-11-24
> **Authors' response**
>
> We thank the reviewer for their review. Please find our answers below:
>
> 1. Performance averaged across training vs final performance:
> We choose the performance averaged across training as the relevant metric as different practitioners may have different computational budgets and therefore train for a different number of training steps. Furthermore, for the same final performance, this also provides a better score to agents that trained faster.
>
> 2. Why did we not consider off-policy methods?
> We were interested in better understanding on-policy methods; hence the focus on these approaches. This already required implementing all the considered implementation choices of on-policy agents in a single unified agent implementation. In contrast, off-policy agents have a very different setup (e.g., with a replay buffer) which would have required a separate unified agent implementation for off-policy agents. While beyond the scope of this paper, we would welcome a similar study for off-policy methods.
>
> 3. Additional topics: "numerical stability of design/hyper-parameter choices, convergence behaviors".
> We agree that these are interesting topics that merit further investigation. Given the already extensive results in this manuscript, we believe that this is best done in separate papers (which this paper could serve as a basis for) focusing on these areas.
>
> 4. Open source implementation:
> We will be open-sourcing our agent implementation.

---

### Public Comment · ~Eugene_Vinitsky1 · 2020-11-13
**Results at convergence**

Thank you for this excellent paper! I just had one point of curiosity, namely, it seems that the scores for some of the environments are significantly lower than I think they would be for a good converged policy. For example, a good policy in hopper will score above 3000 but it seems that your policies score significantly lower than this, likely due to the limited number of training steps. Given this is true (perhaps it is not and there's some difference between your hopper envs and the ones I am used to), do you think that your conclusions still hold for converged policies? Is it possible that suggested hyperparameter choices that appear good for policies that are not fully converged would actually harm the final converged reward? Or is it the case that reward curves do not tend to cross over and so conclusions can be drawn with partially converged rewards. It seems that it would be interesting to take a subset of these policies and give them more training time to see how the conclusions are affected. I'm very curious to know your thoughts on this issue.

Note, if I've misunderstood the reward scale then this question is totally moot.

Thank you!

---

> ### Author Response · Authors · 2020-11-24
> **Authors' response**
>
> Thank you for the comment. Please note that, as described in the paper, the scores are *average* episode returns during the training of *stochastic* policies. Depending on the task, they are either competitive or close to competitive. Moreover, we have verified that the agent which combines the recommendations from all our experiments performs very well in all the environments.

---

### Public Comment · ~Daniele_Reda2 · 2020-11-14
**Complementary work on analysis of environment design choices for RL**

We really appreciate and enjoyed reading this work, as it sheds light on the impact of hyperparameters for on-policy RL algorithms.

We wish to share our concurrent related work, as published earlier this year. It examines the impact of environment design choices for RL. While not formally hyperparameters, these choices impact the learning in a similar manner, and thus understanding these choices is important for RL practitioners.

*“Learning to Locomote: Understanding How Environment Design Matters for Deep Reinforcement Learning”*
Daniele Reda, Tianxin Tao, Michiel Van de Panne
MIG 2020 (ACM SIGGRAPH Conference on Motion, Interaction and Games)
https://www.cs.ubc.ca/~van/papers/2020-MIG-envdesign

It is complementary to this submission in many ways; we examine the impact of:
state representations, initial state distributions, reward structure, control frequency, episode termination procedures, curriculum usage, the action space, and torque limits. Our experiments are performed with TD3 as the RL algorithm of choice, Pybullet as the physics-based simulator, and using common locomotion benchmarks.

As one area of overlap, we also experiment with different ways of handling the termination signal due to timestep limits: in contrast with your findings, our results show that bootstrapping the termination state can improve overall performance. This could imply that the result is either sensitive to the choice of algorithm (on-policy vs off-policy) or the simulator (Mujoco vs PyBullet).

---

### Decision · Program_Chairs · 2021-01-07
**Final Decision**

**Decision:**

Accept (Oral)

**Comment:**

There is a clear consensus over all reviewers that this is a very strong empirical analysis, with actionable insights that should prove quite useful both to researchers and practitioners. I have no doubt that many will use it as a reference when implementing and using RL algorithms (especially since the authors said they would release their code).

This is thus a clear accept, that in my opinion would deserve an oral presentation, so as to better disseminate its key findings.